Nation-scale primary prevention to reduce newly incident adolescent drug use: the issue of lag time

http://orcid.org/0000-0001-7252-5090 Cheng Hui G. 1 chenghu2@msu.edu
Augustin Dukernse 1 2
Glass Eric H. Jr 2
http://orcid.org/0000-0001-7176-0929 Anthony James C. 1 janthony@msu.edu
1 Department of Epidemiology & Biostatistics, Michigan State University , East Lansing, MI , USA
2 Michigan State University , East Lansing, MI , USA
Nock Nora
Electronic publication date: 2019 Feb 12
Publication date: 2019
Volume: 7
Electronic Location ID: e6356
Received 2018 Aug 20; Accepted 2018 Dec 28
Copyright: © 2019 Cheng et al.
Copyright year: 2019
Copyright holder: Cheng et al.
License: This is an open access article distributed under the terms of the Creative Commons Attribution License, which permits unrestricted use, distribution, reproduction and adaptation in any medium and for any purpose provided that it is properly attributed. For attribution, the original author(s), title, publication source (PeerJ) and either DOI or URL of the article must be cited.
License URL: https://creativecommons.org/licenses/by/4.0/

Keywords: Synar amendment, USA, National minimum drinking age act, Incidence

Funding: National Institute of Drug Abuse NIDA T32DA021129 [HGC] and K05DA015799 [JCA] Michigan State University This work was supported by the National Institute of Drug Abuse (NIDA T32DA021129 [HGC] and K05DA015799 [JCA]) and Michigan State University. The funders had no role in study design, data collection and analysis, decision to publish, or preparation of the manuscript.

==============================
Background

There is limited evidence about the effects of United States (US) nation-level policy changes on the incidence of alcohol drinking and tobacco smoking. To investigate the potential primary prevention effects on precocious drug use and to clarify lag-time issues, we estimated incidence rates for specified intervals anticipating and lagging after drug policy enactment. Our hypotheses are (a) reductions in underage drinking or smoking onset and (b) increases of incidence at the legal age (i.e., 21 for drinking and 18 for smoking).

Methods

The study population is 12–23-year-old non-institutionalized US civilian residents. Estimates are from 30 community samples drawn to be nationally representative for the US National Surveys on Drug Use and Health 1979–2015. Estimates were year-by-year annual incidence rates for alcohol drinking and tobacco smoking by 12–23-year-olds, age by age. Meta-regressions estimate age-specific incidence over time.

Results

Incidence of underage alcohol drinking declined and followed a trend line that started before 1984 enactment of the National Minimum Drinking Age Act, but increased drinking incidence for 21 year olds was observed approximately 10 years after policy enactment. Eight years after the Synar amendment enactment, evidence of reduced smoking incidence started to emerge. Among 18 year olds, a slight increase in tobacco smoking incidence occurred about 10 years after the Synar amendment.

Conclusion

Once nation-level policies affecting drug sales to minors are enacted, one might have to wait almost a decade before seeing tangible policy effects on drug use incidence rates.

Introduction

A leading cause of disease burden in young people is precocious and underage use of alcohol, tobacco, and other drugs, by which we mean starting to use one or more of these drugs extra-medically before age 18 years or age 21 years (Hall et al., 2016; Lim et al., 2012). In the United States (US), a series of nation-level policies has been put into place to help shape primary prevention of these forms of precocious drug use, and to deter new onsets of underage drug use. In the case of alcohol, the 1984 National Minimum Drinking Age Act (NMDAA) required all states to ban sales of alcoholic beverages to individuals below age 21 years, and to do so during or before December 31, 1986; otherwise, the state would lose a portion of its federal highway funds (Toomey, Nelson & Lenk, 2009). In the case of tobacco (DiFranza & Dussault, 2005), the 1992 Synar Amendment prohibited sales or distribution of tobacco products to individuals before the 18th birthday. Otherwise, the state would not receive the full award of the federal Substance Abuse Prevention and Treatment Block Grant (https://www.samhsa.gov/synar/about, accessed January 13, 2017).

Faced with challenges of a true experimental design, a series of studies applied novel statistical models to observational data to infer the potential causal role of these policies in alcohol drinking and smoking behaviors (e.g., regression discontinuity and instrumental variable approaches). These studies have consistently documented greater alcohol consumption among 21 year olds compared to 20 year olds, lending support to the idea that minimum legal drinking age can shape drinking behaviors (Carpenter & Dobkin, 2009; Yörük & Yörük, 2013). The evidence about smoking behavior is less clear (Ertan Yörük & Yörük, 2016). With a focus on variations in the population-level incidence rates for alcohol drinking (i.e., the probability of becoming a user) over multiple years, in our own work, we have added new evidence about the peak risk of drinking onset at 21 years of age in the contemporary US. This published evidence is quite consistent with the notion that the minimum legal drinking age plays a role in shaping drinking onset as young people transition from late adolescence to age 21 years (Cheng, Cantave & Anthony, 2016a, 2016b; Cheng, Lopez-Quintero & Anthony, 2017).

Nonetheless, a policy-maker might ask about how quickly a regulatory change in minimum legal age for any given drug would be followed by tangible reductions in population-level incidence rate estimates driven by the individual-level probability of starting to use that drug. This epidemiological question about lag time from policy enactment until an observable policy-preventive effect on population-level drug use incidence rates generally has been mentioned in discussions about methods (Pacula et al., 2014). Indeed, many of the current evaluations of cannabis policy change are based on assumptions about relatively short lag times, and the mixed evidence in the extensive drug policy evaluation literature reflects, to some degree, an expectation of fairly rapid effects of nation-level policies (Fichtenberg & Glantz, 2002; Holford et al., 2014a; Wagenaar & Toomey, 2002). Virtually all of these studies specified a lag time interval at values of 1–5 years. Hence, we were prompted to wonder whether the lag time specification of 1–5 years might be too short when the goal is to see a policy-preventive effect on population-level incidence rates. We speculated that a lag time of 10 years or greater might qualify as a more realistic working hypothesis when the context is that of a relatively large and heterogeneous country such as the United States, and when states might launch initial policy implementation soon after policy enactment, but might not achieve steady state in policy implementation until many years after policy enactment.

Another important issue is confronted when these policy analyses are evaluated from an epidemiological perspective that draws a crisp distinction between prevalence estimates (the probability of being a user) vs. incidence estimates (the probability of becoming a user). In this context, we note that “primary prevention” of drug policies, by definition, should be seeking a reduction in the incidence rate (formation of newly incident users), and might not have any direct secondary prevention effect of shortening duration. Moreover, the prevalence of active drug use is determined in part by carry-forward of drug use that started in prior years and persists into each subsequent year. The incidence rate has no such carry-forward of long-duration drug use. Incidence pertains to the rate at which newly incident drug users are added to the user population, conditional on no prior use. On the basis of general epidemiological principles, we can formulate a working hypothesis that the primary prevention impact of a nation-level policy change, if any, might be seen first in the incidence rate. If nation-level policy affects prevalence but not incidence, this result signals possible policy disruption of on-going drug use, with no primary prevention effects.

With these thoughts in mind, we decided to take an initial empirical step in an epidemiological investigation of the incidence rate evidence about two separate United States (US) nation-level policy changes clearly intended to have beneficial primary prevention effects on precocious or otherwise underage drug use: (1) the NMDAA, and (2) the Synar, as described above. For this initial step, we have produced year-by-year estimates for the annual incidence rates for drug use experiences of 12–21-year-olds, age by age, before and after enactment of these two policies. Our working hypothesis, specified prior to data analysis, was that any primary prevention effect of these nation-level policies on incidence rates might not be seen within a lag time of 5 years. A 10-year lag time might be required.

In order to look into our working hypothesis that it might take 10 years or more to see drug policy impact on adolescent incidence rates, we constructed a US time series of incidence rate estimates for the years before and after the US enacted these two drug policies. A secondary hypothesized change is an increase of incidence at the legal age (i.e., 21 for drinking and 18 for smoking), which is an empirically grounded hypothesis based on some recent work on age-specific incidence rates for starting to drink alcoholic beverages (Cheng, Cantave & Anthony, 2016a). This empirical research report describes what we found.

Materials and Methods

Study population and sample

The study population is non-institutionalized civilian residents aged 12 years and above living in the US, sampled from 30 national surveys conducted by the US federal government. From 1979 to 2001, these surveys were named National Household Survey on Drug Abuse (NHSDA), which changed to National Survey on Drug Use and Health (NSDUH) in 2002 (Substance Abuse and Mental Health Services Administration (SAMHSA), 2014). In all surveys, multi-stage area probability sampling was used to yield US nationally representative samples (Substance Abuse and Mental Health Services Administration (SAMHSA), 2012). More details about sampling procedures and demographic characteristics of the NHSDA and NSDUH surveys are provided on the NSDUH website (https://www.datafiles.samhsa.gov/study-series/national-survey-drug-use-and-health-nsduh-nid13517, accessed November 10, 2018). All NSDUH participants were recruited via child assent and parental or adult consent, based upon protocols approved by cognizant human subjects protection committees.

From 1979 to 1990, the sample size varied from 5,000 to 10,000; between 1991 and 1998, the sample sizes were between 18,000 and 30,000; since 1999, the sample size has been ∼55,000 each year (Substance Abuse and Mental Health Services Administration (SAMHSA), 2012). For this analysis, we focused on the “at risk” population for the onset of alcohol drinking or cigarette smoking. Tables S2 and S3 provides year- and age-specific sample sizes of the “at risk” populations. The aggregated sample sizes for this study are 358,205 for alcohol drinking and 429,348 for tobacco smoking (i.e., denominators for incidence estimates).

Assessment and measures

During NSDUH fieldwork, participants completed interviews, with standardized multi-item modules on alcohol, tobacco, and other health topics. These items assessed the participant’s age when they had their first drink or smoked their first cigarette, as well as recency of use, and lifetime history of use. Minor changes in wording occurred during the 37 years of study interval. Table S1 provides more details about changes in survey questions as well as changes in methodology that are most relevant to this analysis (Substance Abuse and Mental Health Services Administration (SAMHSA), 2014). In this study, newly incident users are individuals whose onset was at the current age or a year younger. This approach yields valid estimates for annual incidence and constrains memory errors (Barondess et al., 2010; Kuntsche et al., 2016; Shillington et al., 2012). Age is based on the date of birth. Information from the survey roster was drawn when the question was skipped.

Analysis approach

In this study, incidence was conceptualized as the number of newly incident users who started drinking or smoking at a specific age rising from the “at risk” population comprised of never users assessed at a specific age and newly incident users. A previous study provides more details about this approach for the estimation of age-specific incidence (Cheng, Lopez-Quintero & Anthony, 2017). From 1999 and onward, the participants’ age is only available in aggregated groups for those older than 21 years in the public downloadable datasets. For example, 22–23 year olds were grouped together, and 24–25 year olds were grouped together. To accommodate this change, newly incident drinkers or smokers at 21 included those who started drinking or smoking at 21 and assessed at 21 and half of those who started drinking or smoking at 21 and assessed at 22 or 23; annual incidence was estimated for 22–23 as a group (instead of for 22 year olds and 23 year olds separately). The incidence estimate was operationalized as analysis-weighted numbers of newly incident users divided by analysis-weighted numbers of those at risk to start using. In subsequent steps, we plotted the estimated annual incidence for visualization of change over time.

Next, we used meta-regression to compare incidence estimates between four time periods based on the NMDAA and Synar amendments (Thompson & Sharp, 1999). Conceptually, meta-regression aggregates evidence to address sources of heterogeneity in related estimates from multiple studies on the same topic. with each study viewed as a non-exact replicate based on different study populations across various spatio-temporal locations (Borenstein et al., 2009). Meta-regression takes into account sampling variability (via error terms) and study-specific deviations from the general mean (via random effect terms) in order to quantify estimated heterogeneity in estimates associated with study-specific characteristics (e.g., study period). In this research, the annual NSDUH surveys over 37 years are the “multiple studies,” each with an independent replicate sample of the US study population as it changed year by year, and with methods changes as noted in Table S1, and the estimands are incidence rate estimates. One might use other analysis approaches, but our pre-specification for the analysis protocol was to make use of meta-regressions and to report what we learned from meta-regressions. Our data sharing plan makes it possible for others to use different approaches and to learn whether different approaches might yield different results. For alcohol drinking, the four periods are 1979–1986 (pre-NMDAA, Period 0), 1987–1996 (immediate post-NMDAA, Period 1), 1997–2006 (post-NMDAA, Period 2), and 2007–2015 (post-post-NMDAA, Period 3). For tobacco smoking, the four periods are 1979–1992 (pre-Synar, Period 0), 1993–1999 (immediate post-Synar, Period 1), 2000–2007 (post-Synar, Period 2), and 2008–2015 (post-post-Synar, Period 3).

Analysis weights account for sample selection probabilities and post-stratification adjustment factors that yield US Census subpopulation counts.

Results

Figure 1 depicts a visualization of change over time for each age stratum. Table S3 shows estimated age-specific annual incidence of drinking among 12–23 year olds, year by year, from 1979 to 2015. Among adolescents 12–17 years of age, sharp declines occurred before NMDAA and continued into the early 1990s, paused during the 1990s, and a second wave of declines with smaller magnitude occurred afterward. Consistent with our hypothesis, annual incidence of drinking among 21 year olds rose; however, the increase was not observed until 1991 (i.e., 7 years after NMDAA). Among 18–20 year olds, the largest decrease occurred before the NMDAA (i.e., between 1979 and 1982). The annual incidence of drinking remained fairly stable at 40% for 18 year olds, whereas the decline continued for 19-and-20-year-olds during the 10 years after NMDAA (i.e., 1985–1994). Since 1998, drinking incidence has been fairly stable among 18–20 year olds. In 22–23 year olds, annual incidence of alcohol drinking remained at a low level (i.e., <10% except for 1979).

Figure 1 Estimated annual incidence (%) of alcohol drinking in the United States among 12–23 year olds from 1979 to 2015.

Data from the United States National Surveys on Drug Use and Health, 1979–2015 (n = 358,205). (A) 12–14 year olds. (B) 15–17 year olds. (C)18–20 year olds. (D) 20–23 year olds.

As shown in Fig. 2 and Table S4, there was a universal rise in the annual incidence of tobacco cigarette smoking immediately after the Synar amendment in all age groups except for 22–23 year olds between 1993 and 1998, followed by a declining trend that persisted to the most recent survey year among adolescents. Sharper declines were seen for early adolescents compared to later adolescents. The decline occurred earlier among 12 year olds in 1995. Among 18 year olds, no decline was observed. Instead, an ascendance is seen between 2005 and 2010 (i.e., 13 years after Synar). Among 19–23 year olds, the incidence of tobacco cigarette smoking fluctuated without any clear time trend since the late 90s.

Figure 2 Estimated annual incidence (%) of tobacco cigarette smoking in the United States among 12–23 year olds from 1979 to 2015.

Data from the United States NSDUH, 1979–2015 (n = 429,348). (A) 12–14 year olds. (B) 15–17 year olds. (C) 17–19 year olds. (D) 20–23 year olds.

Findings from meta-regression disclosed similar patterns as described above (Table 1). Major drops among 12–20 year olds occurred during the first post-NMDAA period, stopped during the second decade after NMDAA, and resumed with a lesser magnitude during the third decade after NMDAA. Increases among 21 year olds occurred in the second decade and continued into the third decade after NMDAA.

Table 1 Estimated difference in incidence (% and 95% CI) of alcohol drinking and tobacco cigarette smoking across different time periods based on the National Minimum Drinking Age Act and the Synar Amendment.

Age	Period 3 vs. Period 2	Period 3 vs. Period 1	Period 2 vs. Period 1	Period 3 vs. Period 0	Period 2 vs. Period 0	Period 1 vs. Period 0	
Alcohol	
12	−3.6 (−5.2,−2.0)	−3.9 (−5.8,−1.9)	−0.3 (−2.2,1.7)	−15.9 (−20.9,−10.9)	−12.3 (−17.3,−7.3)	−12.1 (−17.2,−6.9)	
13	−4.7 (−6.3,−3.1)	−5.5 (−7.6,−3.4)	−0.8 (−2.9,1.4)	−19.2 (−23.5,−14.8)	−14.4 (−18.8,−10.1)	−13.7 (−18.3,−9.1)	
14	−5.0 (−7.3,−2.7)	−4.3 (−7.0,−1.5)	0.7 (−2.0,3.5)	−14.1 (−19.9,−8.4)	−9.1 (−14.8,−3.4)	−9.9 (−15.8,−3.9)	
15	−4.4 (−6.5,−2.3)	−2.1 (−4.9,0.7)	2.3 (−0.5,5.1)	−22.6 (−28.8,−16.3)	−18.1 (−24.4,−11.9)	−20.5 (−27.0,−13.9)	
16	−2.4 (−4.2,−0.6)	−3.4 (−6.0,−0.9)	−1.0 (−3.6,1.6)	−17.9 (−24.0,−11.9)	−15.5 (−21.5,−9.5)	−14.5 (−20.8,−8.2)	
17	−2.6 (−4.5,−0.6)	−1.3 (−4.6,2.0)	1.3 (−2.1,4.6)	−18.3 (−26.0,−10.5)	−15.7 (−23.5,−8.0)	−17.0 (−25.2,−8.8)	
18	1.1 (−1.7,3.9)	4.7 (0.1,9.3)	3.6 (−1.0,8.2)	−14.7 (−25.6,−3.7)	−15.8 (−26.8,−4.9)	−19.4 (−30.9,−7.9)	
19	−2.0 (−4.7,0.6)	2.1 (−2.3,6.4)	4.1 (−0.3,8.5)	−22.6 (−32.6,−12.6)	−20.6 (−30.6,−10.6)	−24.7 (−35.3,−14.1)	
20	0.4 (−2.6,3.4)	6.4 (1.9,10.9)	6.0 (1.4,10.6)	−12.4 (−26.0,1.2)	−12.8 (−26.4,0.8)	−18.8 (−32.8,−4.8)	
21	13.1 (8.4,17.8)	27.3 (20.4,34.2)	14.2 (7.2,21.1)	29.4 (16.1,42.7)	16.3 (2.9,29.7)	2.1 (−12.2,16.4)	
22–23	1.8 (0.6,2.9)	3.8 (2.3,5.2)	2.0 (0.6,3.4)	−0.4 (−5.7,5.0)	−2.1 (−7.5,3.2)	−4.1 (−9.6,1.3)	
Tobacco	
12	−3.1 (−4.3,−1.9)	−8.0 (−9.5,−6.4)	−4.8 (−6.4,−3.3)	−8.1 (−10.0,−6.1)	−4.9 (−6.9,−3.0)	−0.1 (−2.3,2.1)	
13	−3.9 (−6.3,−1.6)	−9.6 (−12.2,−7.0)	−5.7 (−8.1,−3.2)	−8.2 (−11.2,−5.3)	−4.3 (−7.1,−1.4)	1.4 (−1.7,4.5)	
14	−4.1 (−6.5,−1.7)	−9.5 (−12.2,−6.7)	−5.4 (−8.0,−2.8)	−5.1 (−8.1,−2.1)	−1.0 (−3.9,1.9)	4.4 (1.2,7.5)	
15	−3.7 (−5.9,−1.5)	−7.7 (−10.3,−5.1)	−4.0 (−6.5,−1.5)	−3.5 (−6.5,−0.4)	0.2 (−2.7,3.1)	4.2 (1.0,7.5)	
16	−3.3 (−4.8,−1.8)	−6.3 (−8.4,−4.3)	−3.0 (−5.0,−1.0)	−0.7 (−3.0,1.6)	2.6 (0.3,4.8)	5.6 (3.0,8.2)	
17	−0.9 (−2.6,0.8)	−4.5 (−6.9,−2.2)	−3.6 (−5.9,−1.3)	1.8 (−1.1,4.8)	2.7 (−0.2,5.7)	6.3 (3.0,9.7)	
18	2.4 (−0.4,5.1)	3.8 (0.5,7.1)	1.4 (−1.7,4.5)	12.0 (8.5,15.4)	9.6 (6.3,12.9)	8.2 (4.4,11.9)	
19	1.8 (>−0.1,3.6)	1.3 (−1.2,3.8)	−0.5 (−2.9,1.9)	5.1 (2.0,8.1)	3.3 (0.3,6.2)	3.8 (0.3,7.2)	
20	1.6 (0.1,3.2)	1.1 (−1.0,3.1)	−0.6 (−2.4,1.3)	4.1 (1.6,6.6)	2.4 (0.0,4.8)	3.0 (0.3,5.7)	
21	−0.1 (−1.7,1.5)	−0.7 (−2.9,1.5)	−0.6 (−2.7,1.4)	4.5 (2.3,6.7)	4.6 (2.5,6.7)	5.2 (2.7,7.8)	
22–23	0.2 (−0.2,0.7)	0.6 (0.1,1.1)	0.3 (−0.2,0.8)	1.4 (0.8,1.9)	1.1 (0.6,1.6)	0.8 (0.2,1.3)	
Notes:

Data from NSDUH, 1979–2015.

For alcohol drinking, Period 0 = 1979–1986, Period 1 = 1987–1996, Period 2 = 1997–2006, Period 3 = 2007–2015; for tobacco smoking, Period 0 = 1979–1992, Period 1 = 1993–1999, Period 2 = 2000–2007, Period 3 = 2008–2015. Bold font indicates statistical significance at 0.05 level.

For tobacco smoking, the first 7 years after the Synar amendment saw universal increase in smoking incidence among all age groups except for 12 and 13 year olds (Table S4). Compared to the first post-Synar period (i.e., 1993–1999), tobacco smoking incidence during the second post-Synar interval declined for all adolescents and persisted to the most recent period under study. In contrast, incidence of tobacco smoking increased slightly among 18 year olds, and remained stable for 19–23 year olds during the second and third post-Synar periods.

Readers may be interested in assessing the shift of age patterns over time. Figure S1 presents age-specific patterns for alcohol drinking incidence. Before 1986, drinking incidence was high among 15–20 year olds. From 1988 to 1994, the peak was generally observed between 16 and 18, and a secondary peak at 21 gradually emerged. From 1995 and onward, the peak at 21 became prominent and the incidence curve shifted from a bi-modular shape to a uni-modular shape. A noticeable dip at 19 and 20 years was seen in 1993 for the first time and became consistent since 1998.

Figure S2 shows the age pattern for tobacco cigarette smoking year by year. Before 1993, there were no appreciable differences across age strata among 12–21 year olds. From 1993 to 1997, a drop after 18 years emerged. Closer scrutiny suggests that the drop was mainly due to a rise among 15–18 year olds. Before 2002, there is no specific peak; incidence was noteworthy among 15–18 year olds. Starting at 2002, a peak at 18 years started to form and became prominent after 2004.

Discussion

The main findings of this study may be summarized succinctly. First, declines in incidence of adolescent alcohol drinking after NMDAA appear as a continuation of a decreasing trend that started before the NMDAA. The rise among 21 year olds did not emerge until approximately 10 years after NMDAA. Two decades after NMDAA, a second decline occurred among adolescents. Second, the annual incidence of tobacco cigarette smoking rose immediately after the Synar amendment in all age groups. Declines in adolescent tobacco smoking incidence occurred at least 8 years after the Synar amendment and persisted until the most recent year under study. Among 18 year olds, a slight increase in tobacco smoking incidence occurred about 10 years after the Synar amendment and lasted for about 10 years. Third, the shift of age pattern in incidence occurred gradually. Prominent onset peaks of autonomous drinking or smoking at the legal age did not occur until approximately 10 years after the NMDAA or Synar.

Before detailed discussion of these results, several of the more important study limitations merit attention. Of central concern is the observational nature of the study which precludes definitive evidence for any causal relationships. Many changes related to alcohol drinking and tobacco smoking co-occurred with NMDAA and Synar. For example, there were numerous nation-, state-, and local-level policies and movements, including excise taxes on tobacco and alcohol products, state-level MLDA or minimum legal tobacco purchasing age, mass media campaigns, and grassroots movements throughout the study period (Farrelly, Niederdeppe & Yarsevich, 2003; Lum, Barnes & Glantz, 2009; Toomey, Nelson & Lenk, 2009), as well as variations in the implementation of state-level policies over time (Nelson et al., 2015; Spivak & Monnat, 2015). The observed changes are results of the national NMDAA and Synar combined with these various factors.

We wish to return to the issue of unmeasured and uncontrolled variables in our meta-regressions. We are hopeful that readers will appreciate that this is an observational study based on a national-level surveillance of alcohol and tobacco use in the US population over a 37 year interval. Nation-level surveillance of this type cannot cover all potentially influential variables that might show variation within or across US jurisdictions during a 37-year interval, but should not be subject to influence from relatively stable variables such as the ratio of males and females among newborns in the US population. Nevertheless, there are some variables that have changed during this 37-year interval and that might have a bearing on the estimation of lag times from the year of policy enactment until a tangible reduction in drinking or smoking incidence rates can be seen. We must leave the control of these variables to a future research project that can build from our meta-regression findings, and in this set of variables a future research team will face some special challenges that involve topics such as disposable income and local alcohol/tobacco taxes, the smoking behavior of parents or siblings or peers or adult co-workers, and the relative balance of US-born vs. non-US-born residents. A participant’s disposable income is not a variable that is measured in these surveys, nor is the local area or state alcohol or tobacco tax level as it has varied over time. A measure of parental or other environmental alcohol and tobacco consumption can be created and introduced as a meta-regression covariate, but we have not done so in this initial meta-regression work. The same is true for the relative balance of US-born vs. non-US-born residents of the US, given that the recent US population distributions show a reduced proportion of US-born residents, relative to what was seen in the first 10 years of the 37-year interval under study. We are not at all sure that changing variables of this type would have a major impact on the meta-regression estimates about lag time for policy effects on population-level incidence rates, and we must leave these issues as ones that can be addressed in future investigations that build from the present investigation.

There are several significant methodological changes during the 37 years under study. First, the NSDUH sampling frame changed in 1991 to include Alaska and Hawaii, as well as civilians living in non-institutional group quarters (Substance Abuse and Mental Health Services Administration (SAMHSA), 2014). Nonetheless, neither NMDAA nor Synar happened in 1991, and no out-of-line estimates were observed in 1991 for either alcohol drinking or tobacco smoking. Second, the sample sizes of the earliest surveys are relatively small which constrains the power of statistical inference. Third, the time span of the current analysis is restricted to 1979 and later due to the availability of data. Therefore, we were not able to assess trends in alcohol drinking for an extended pre-NMDAA period. Fourth, the survey questions related to alcohol drinking and tobacco smoking changed several times. Of special interest is the change of smoking question in 1994, which might have contributed to the increase in tobacco incidence. Nonetheless, the increase is observed in other population surveys for which survey questions did not change in 1994 (e.g., MTF and National Youth Risk Behavior Survey; Johnston et al., 2015; Centers for Disease Control and Prevention (CDC), 2013). Therefore, we do not consider the increase an artifact wholly attributable to change in survey questions. In addition, the method of survey administration changed from face-to-face interview to ACASI in 1999. Nonetheless, no discernible change was observed around 1999. Although several method changes, including the provision of a $30 incentive, occurred in 2002 and the survey name changed, questions about the onset of drinking and smoking as well as the sampling strategy remained the same. In addition, no substantial changes in estimated incidence of drinking or smoking was observed. Therefore, we consider that these changes had minimum influence on our estimates for this study.

Although the assessment of first full drink is fairly straightforward, the first use of tobacco product is less so. Other forms of tobacco products exist, including chewing tobacco, water pipe smoking, and recently vaping (i.e., e-cigarettes). In this study, we only focused on the first tobacco cigarette smoking in order to generate comparable estimates. (The use of other forms of tobacco products was not assessed in earlier surveys.) According to the NSDUH data, approximately 6–10% of all tobacco users had never smoked tobacco cigarettes between 2002 and 2015. (NSDUH has not been assessing e-cigarette use.) Given the relatively small proportions, we judge that the use of other tobacco products cannot fully account for the observed changes over time.

Notwithstanding limitations such as these, the study findings are of interest because we provided an over-time view of age-specific changes in the onset of drinking and smoking using nationally representative data with fairly consistent methodology. For this study, we specified the study population to include essentially all non-institutionalized civilian young people in the US, even if the young people had dropped out of school or had become disengaged from school, an experience that often signals excess incidence rates of drug use (Sweeten, Bushway & Paternoster, 2009). In this fashion, we avoided potential study biases faced when nation-level policy has been evaluated based on school survey data.

This study population feature of the research design might be especially important because during the years under study there were concurrent initiatives to expel, suspend, or restrict school attendance of students who had started to smoke tobacco or to drink alcoholic beverages (DiFranza & Dussault, 2005; Toomey, Nelson & Lenk, 2009). In consequence, corresponding incidence rates for school-attending adolescents would not provide a complete picture of the epidemiological phenomenon. In addition, the available school surveys generally have asked about “grade” of starting drug use, but not “age” of starting drug use, with “grade” being undefined for one who no longer attends school. The NMDAA and Synar policies focus on age of starting to use, but not on grade of starting to use. Moreover, our intent was to reach out and to capture the experience of 18–21-year-olds, many of whom are not included in the sampling frames for US surveillance of school-attending youths, year by year, from the late 1970s through the 2000s.

In this study, we found large drops in adolescent drinking incidence before and after the NMDAA. It is noteworthy that NMDAA is not the first to impose age 21 as the legal minimum drinking age (LMDA). In 1933 (after the nation-wide prohibition was removed), a LMDA of 21 was enacted in all states until 1970. Between 1970 and 1975, 29 states lowered the LMDA to 18. Between 1978 and 1986, these states reinstated the LMDA to 21 due to concerns about a rise of motor vehicle accidents among 18–20-year-olds (Toomey, Nelson & Lenk, 2009). The implementation of NMDAA happened during 1978–1986 in different states. Therefore, whether the decline in adolescent drinking onset is at least partially due to NMDAA is not clear. Nonetheless, the distinctive peak at 21 was not seen until some 10 years after NMDAA.

For tobacco smoking, even though the Synar amendment was the first nation-level policy to set an age limit on sales and distribution of tobacco products to minors, the 1964 Surgeon General’s report was a milestone that brought an array of subsequent changes against tobacco smoking including high excise taxes, banning of tobacco advertising to youth, and banning of tobacco smoking in public places (Gruber, 2001). Previous evidence with a relatively short time span did not support any immediate effect of Synar on smoking prevalence in adolescents (Fichtenberg & Glantz, 2002). In this study with more than 20 years of elapsed time since Synar, we found drops in tobacco cigarette smoking incidence ∼10 years after the Synar amendment. Larger drops were seen for young adolescents than older adolescents.

It is worth noting that there were increases in smoking incidence in the mid-90s in all age groups, signaling robust efforts to promote smoking for the general population. Examples of such effort include significant drops of price and marked increase in advertising and marketing activities in the early 1990s (Chaloupka & Wechsler, 1997; Emery, White & Pierce, 2001; Fichtenberg & Glantz, 2002; Gruber, 2001; Gruber & Zinman, 1997; Holford et al., 2014b; Lum, Barnes & Glantz, 2009; Siegel et al., 1997; National Center for Chronic Disease Prevention and Health Promotion (US) Office on Smoking and Health, 2012; World Health Organization (WHO), 2015). Other potential influences include the introduction of nicotine patch in the early 90s, which can provide a false promise for easy cessation, as well as increased cannabis smoking in the mid-90s (Sugawara & Nikaido, 2014) and potential secular changes in individual-level characteristics (e.g., risk-taking behaviors). Despite these co-occurring changes, tobacco smoking incidence in adults 18 years of age and older did not change appreciably after the first 8 years post-Synar, but decreased among adolescents. Less clear is whether the promotion activities for tobacco smoking canceled out any effect of Synar in the mid-90s.

Summarizing from our findings for tobacco and alcohol, there does not seem to have been any instant effect of NMDAA and Synar on the onset of drinking or smoking; it took ∼10 years before any observed variation at the population level (i.e., delays of onset and peaks at the legal age). Nonetheless, any effect of such policies might be a long lasting one, as evidenced by the declining trend of adolescent drinking and smoking that continued into the second decade after NMDAA and Synar.

The results from this study may have important implications in future efforts to seek empirical evidence about potential effects of policy changes on psychoactive drugs, including evidence about cannabis during an era of cannabis policy change (Pacula et al., 2014). Based upon findings of this study, no significant shift in the incidence curve for cannabis onset would occur during the first 10 years after the implementation of cannabis policies. This may explain the mixed evidence we have seen for cannabis use; that is, the mixed evidence can be a mere reflection of sampling variation when there has been no real change (Azofeifa et al., 2016; Cerdá et al., 2012; Choo et al., 2014; Harper, Strumpf & Kaufman, 2012; Hasin et al., 2015; Keyes et al., 2016; Lynne-Landsman, Livingston & Wagenaar, 2013; Schuermeyer et al., 2014; Wall et al., 2011). Close monitoring, with special focus on the timing of onsets of youth cannabis use, might provide more definitive evidence about the lag time for cannabis. Nonetheless, local-level (e.g., school-based) prevention initiatives might have a more immediate effect on adolescent cannabis onset, and these initiatives might well be needed during the first 10 years after any nation-scale policy change in order to reduce youth cannabis onset and to prevent unnecessary adverse consequences of early onset of cannabis use, which might include cognitive impairments, drugged driving, and dropping out of school (Pacula et al., 2014).

Supplemental Information

Supplemental Information 1 This file provides supplementary material mentioned in the main text.

Click here for additional data file.

The authors are grateful to the United States Department of Health and Human Services, Substance Abuse and Mental Health Services Administration for making the data publicly available. Authors are grateful for valuable advice from Dr. Carla Storr.

Additional Information and Declarations

Competing Interests

Author Contributions

Data Availability

The authors declare that they have no competing interests.

Hui G. Cheng conceived and designed the experiments, performed the experiments, analyzed the data, contributed reagents/materials/analysis tools, prepared figures and/or tables, authored or reviewed drafts of the paper, approved the final draft.

Dukernse Augustin performed the experiments, analyzed the data, contributed reagents/materials/analysis tools, approved the final draft.

Eric H. Glass Jr performed the experiments, analyzed the data, contributed reagents/materials/analysis tools, approved the final draft.

James C. Anthony conceived and designed the experiments, contributed reagents/materials/analysis tools, authored or reviewed drafts of the paper, approved the final draft.

The following information was supplied regarding data availability:

The datasets (1979–2015) were downloaded from the National Survey on Drug Use and Health (NSDUH): https://www.datafiles.samhsa.gov/study-series/national-survey-drug-use-and-health-nsduh-nid13517.

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
