# Peer review of "Nation-scale primary prevention to reduce newly incident adolescent drug use: the issue of lag time"

_PeerJ, doi:10.7717/peerj.6356_

## Round 0.1 · original submission · Major Revisions

Your manuscript has been reviewed and found to require major revisions before further consideration. Please provide a point-by-point response discussing how and where in the revised manuscript each issue raised was addressed.

Reviewer 1 ·

Basic reporting

no comment

Experimental design

no comment

Validity of the findings

no comment

Additional comments

This is a well written paper that examines whether there is a lagged effect of important alcohol and smoking control policies, the Minimum Legal Drinking Age and Minimum tobacco purchasing age on drinking and smoking behavior. They document that the increase of drinking incidence among 21 year olds was not seen until approximately 10 years after the policy change. Declines in adolescent tobacco smoking incidence occurred at least eight years after the Synar amendment. I recommend a major revision and hope that the authors will successfully address the methodological problems that I address below.

The abstract of the paper is difficult to follow for the general audience. In the methods section the author may want discuss that they use the Meta analysis. In their methodology section, they should discuss why the meta analysis is the best practice to measure the lagged effect of these laws.

The Authors claim that “ The effects of United States (US) nation-level policy changes on the incidence of alcohol drinking and tobacco smoking are not fully understood. “ However, there is an extensive literature on the impacts of alcohol prevention policies on alcohol related outcomes. Furthermore, the effects of tobacco policies have also been discussed. (Wagenear and Toomey (2002), Carpenter et al. (2007))

The authors report that the increase of drinking incidence among 21 year olds was not seen until approximately 10 years after the policy change. There does not seem to have been any 346 instant effect of NMDAA and Synar Act on the onset of drinking or smoking. However, there are large number of studies that show the immediate effect of the minimum legal drinking age on several alcohol related outcomes. When the young adults turn 21, there is a sharp increase in the probability of drinking. (See for example, Carpenter and Dobkin (2009)

The authors may also investigate the effects of these laws especially the minimum legal drinking age (MLDA) and minimum legal tobacco (MLTPA) purchasing age on young adults who belong to different demographic groups or who are subject to different MLDAs and MLTPAs such as 18 and 19.

The majority of states in the United States requires that a couple be 18 in order to marry without parental permission. Marital status may significantly affect smoking behavior of youths. It is not clear how the authors address this problem.

·

Basic reporting

IMPORTANT ACTIONS:
i. Restructure text to provide focus to the introduction and discussion as highlighted below
ii. Reconsider Tables and Figures 1 and 2 – do Tables/Figures provide different messages? Do either enable a focussed insight into the research question?
iii. Address demographics of sample

Further comment:

‘Uptick’ not in common usage outside the US. Reconsider?

Introduction lines 114 – 130 better suited in limitations and strengths discussion, by repositioning more emphasis would be put on the research question. 324-330 is main finding but somewhat buried in the discussion, better placed up front in discussion.

Methods could mention the source of the NSDUH data – SAMHDA. Authors indicate sample size ranges in different years, though stipulating the total sample size would also be beneficial to understand the scale of the work (as provided in table and figure headings).

Some demographic information would be beneficial to the manuscript, such as gender and income group. If not available, then clarification over the sampling method NSDUH employs would provide confidence that demographic bias is minimised.

Figures have been generated with quality statistical software
- It isn’t made clear what the vertical lines in graph indicates, though it is assumed to be the year of the policy. Annotation would clarify.
Tables are comprehensive, but do not particularly offer an interpretable outcome.
- Im not certain what message Tables 1 and 2 offer that Figures 1 and 2 do not provide. As the figures enabling an immediate interpretation of results, the tables could be supplementary.
- I would like to see clarification of the parenthesised numbers. I assume these to be 95%CI of the sample proportion reflecting the true population, but this isn’t defined. Similarly 95%CI for mean difference should be stipulated.
- Table 3 shows many differences with periods 1-3 compared to period 0. However we know there are sample size issues with period 0.
o Can periods be given more intuitive names as defined in the methods. As Periods 0-3 are repeated for alcohol and tobacco, but the years subtly change, it would be clearer if the descriptive name was provided.
o It seems the real crux of the research question rests in table 3 comparing period 3 to either period 2 or 1. The importance of this comparison, in context of the hypothesis, is lost in the complex table.
o Summative figure would help highlight the differences according to time relative to policy, rather than the granular detail of age.
- Whilst the authors mention a second hypothesis around a rise in incidence at age 21, no figure or table specifically addresses this (though apparent in findings). Address or remove to avoid confusion, can be discussion point.

Experimental design

IMPORTANT ACTIONS
i. Results need to focus on research question rather than granularity by age
ii. Methods have minimal statistical analysis, use more advanced approaches

Further comment:

Research question at times a bit convoluted, but interesting, important and study approach addresses gap in knowledge.
- Question seemed to focus on adolescents, and whether there was a change in incidence 10 years post policy. However the granular results make it difficult to see wider patterns. Whilst sub/sensitivity analyses could address whether there are differences between e.g. 12, 13, 14 year olds (as focussed on here), it seems this was not the main intention of the research question, and pooling age data (even over more intuitive age groups? 12-14, 15-17, 18-21, 22+) despites its limitations, may allow a more focussed statistical analysis of the change in incidence over time. The authors already take an approach to combine data in periods of study in relation to the policy implementation.

Large datasets analysed in a simple manner
- Analysis feels unfocussed, matrices of proportions are presented. Results seem unfinished, those presented do not wholly address the research question(s) in a way that tests the hypothesis laid out in the introduction.
- I am unsure why the authors have not performed a form of time series regression, such as interrupted time series, which would address the impact of the policy intervention. Regression analyses would also allow adjustment for age, enabling a clearer message. A statistical approach beyond a descriptive comparison of proportions would be appropriate.
- Rather than a focus on the granularity of the data, the authors could compare variation in annual incidence (all years, or periods) between the ages – it appears younger ages would show less variation whilst older teens would show more variation. Steps should be taken to ensure clear messages emerge from the results, requiring minimal deciphering.
- Previous publications from the authors show robust statistical approaches to derive a clear message, including meta-analyses. Similar approach needed.

Table 3 comes the closest to addressing the research question posed in the introduction, however this could be refined and built upon. Considering the focus of 10 years, compared to studies that look earlier, the periods defined in table 3 appear a pertinent foundation to continue the analysis. Results presented prior to table 3 are informative description of the dataset, yet somewhat peripheral, taking focus away from the objective of the research.

Validity of the findings

IMPORTANT ACTIONS
i. Address NSDUH statement that 2002+ data should not be compared to earlier data for differences across time.
ii. Refocus the manuscript and results on the research question so the added value compared to previous publications is evident. Avoid needless replication of published findings.
iii. Address limitations as outlined below

Further comment:

- The authors have recently published a number of manuscripts using the data source from 2002 onwards. The authors clearly have expertise with the data set. I am concerned that the secondary hypothesis defined in the introduction has already been answered and published by the same authors: although I do not have access, the abstract would suggest so
- “Looking across age strata, we found rising age-specific drinking incidence rates across adolescence to a plateau at age 16–18 years and made a new discovery of a statistically robust and highly reproducible dip in incidence at age 19–20 years, followed by the major peak at age 21 years, with sharply reduced incidence thereafter.” https://doi.org/10.15288/jsad.2016.77.405
- I am also unsure how the tables presented here offer value further to the above publication (other than addition of tobacco results).
- The hypothesis focuses on time-lag, and I feel the authors should focus their analysis on the time-lag in order to offer new insight. I don’t feel that Table 3 adequately supports or rejects the hypothesis, but makes a good foundation for further figures and analysis.

The authors have nobly identified limitations, though emphasis could be redirected.
- Observational study (no causation) – good, but could other techniques identify a strength of an association that offers more grounds for causation?
- Various multi-level policies in addition to NMDAA and Synar. – Yes, relevant.
- Methodological changes in collection pre1991 – Perhaps more emphasis needed on the sampling methods pre2001
- Small sample sizes in earlier years – Yes, well stated but contextualise, could this over or underestimate the differences according to periods?
- Restricted years from 1979 onwards – Is this important? more methodological than discussion, and sample size was low pre1990 anyway, would extension to 1970 or 1960 add further value?
- Survey question change in 1994 – you have evidence in the literature that suggests you can reduce the emphasis of this limitation
- First tobacco cigarette smoking – this is an excellent point and demonstrates a strength of the study, consistent approach. Perhaps more could be stated about the potential impact of e-cigarettes on incidence, how this is recorded, or how future study should address it?

Limitations i would like to see addressed
- How weighting in the survey may influence sample, or whether unweighted data needs to be adjusted with reference to education, deprivation.
- How sample bias may influence sample, NSDUH incentive scheme of $30
- “Therefore the data from 2002 and later should not be compared with data collected in 2001 or earlier to assess changes over time.” Notes on NSDUH site provided in supplementary. The strategy appears to ignore this issue, discussion of its limitation should be much more transparent.

Discussion has addressed possible reasons for increases in tobacco incidence following policy. Speculation could be offered with regard to reasons for a lag, particularly as the manuscript title and research question focus on this. The relevance of adolescent risk-taking behaviour to the results could also be described here.

Additional comments

Thank you for the opportunity to review your manuscript. I take a consistent approach to reviewing all manuscripts that I accept, and offer a full critique. The length of the review is not a reflection of the quality of the manuscript; I always provide what I hope to be sufficient feedback for authors to address and enhance their manuscript. I feel it fair that you receive in-depth feedback to acknowledge the time, money and effort invested in your work.

---

## Round 0.2 · accepted · Accept

The revised manuscript is much improved and all issues raised by reviewers have been sufficiently addressed.

Reviewer 1 ·

Basic reporting

No comment

Experimental design

no comment

Validity of the findings

no comment

·

Basic reporting

No comment

Experimental design

No comment

Validity of the findings

No comment

Additional comments

Thank you for your responses, the manuscript is a well presented study that clearly defines its research goal; the methodology you have taken (for others to reproduce or alternate) and the limitations are very well outlined. The changes made offer focus on the take home message, which adds value to the community. This manuscript now offers excellent foundations for further studies into policy changes and lag-time, highlighting the possible issues with previous efforts and examples of what must be overcome in future projects. You have considered your response appropriately and have produced an insightful report. Good work.